# Comprehensive Analytical Modelling of an Absolute pH Sensor

**DOI:** 10.3390/s21155190

**Published:** 2021-07-30

**Authors:** Cristina Medina-Bailon, Naveen Kumar, Rakshita Pritam Singh Dhar, Ilina Todorova, Damien Lenoble, Vihar P. Georgiev, César Pascual García

**Affiliations:** 1Device Modelling Group, School of Engineering, University of Glasgow, Glasgow G12 8LT, UK; Naveen.Kumar@glasgow.ac.uk (N.K.); r.dhar.1@research.gla.ac.uk (R.P.S.D.); 2326960T@student.gla.ac.uk (I.T.); 2Nano-Enabled Medicine and Cosmetics Group, Materials Research and Technology Department, Luxembourg Institute of Science and Technology (LIST), L-4362 Esch-sur-Alzette, Luxembourg; damien.lenoble@list.lu

**Keywords:** nano-biosensor, analytical model, oxide degradation, depletion width, pH sensor modelling and simulations

## Abstract

In this work, we present a comprehensive analytical model and results for an absolute pH sensor. Our work aims to address critical scientific issues such as: (1) the impact of the oxide degradation (sensing interface deterioration) on the sensor’s performance and (2) how to achieve a measurement of the absolute ion activity. The methods described here are based on analytical equations which we have derived and implemented in MATLAB code to execute the numerical experiments. The main results of our work show that the depletion width of the sensors is strongly influenced by the pH and the variations of the same depletion width as a function of the pH is significantly smaller for hafnium dioxide in comparison to silicon dioxide. We propose a method to determine the absolute pH using a dual capacitance system, which can be mapped to unequivocally determine the acidity. We compare the impact of degradation in two materials: SiO${}_{2}$ and HfO${}_{2}$, and we illustrate the acidity determination with the functioning of a dual device with SiO${}_{2}$.

## 1. Introduction

Ion-Sensitive Field Effect Transistors (ISFETs) [1,2,3] are devices that measure acidity, which offer the best accuracy and miniaturisation. They employ semiconductor fabrication techniques that lower the cost per sensor while providing a high level of sophistication by the integration of the sensors with metal-oxide-semiconductor (MOS) circuits and microfluidics [4,5]. ISFETs transduce the ion activity in the solvent by a capacitance effect that measures the associate charge. Ions are adsorbed from the electrolyte depending on the bulk concentration and the affinity of the material interface. The final signal is formed with the contribution of the charged particles in solution that react forming the double layer capacitance [6,7,8]. There are numerous applications of ISFETs described in the literature ranging from sweat biomarker sensors, physiological measurements, to monitoring the enzymatic activity of polymerase to assist DNA sequencing [9,10,11,12].

Regardless of the impressive progress, these devices have triggered a serious limitation known to anyone using ISFETs, which is that they require continuous re-calibration. ISFETs show an instability that can be manifested in the drift of the threshold voltage or the output current used to transduce the acidity/basicity [13,14,15]. This effect limits the applications that require accurate monitoring of the pH during periods lasting hours as well as the miniaturisation of highly multiplexed devices. As a result of the drift, ISFETs require compensation strategies based on calculations or using reference devices that require extra resources.

At the origin of the drift, there is an irreversible chemical degradation of the dielectric barrier responsible for the change in capacitance. There are several explanations proposed for this effect that involve the migration of charges into the oxide materials [16,17,18,19] that decreases the dielectric constant of the affected region [20]. When possible, the re-calibrations are achieved using external reference buffers with known acidity. More sophisticated systems make use of an internal generation of acid that performs a titration curve [21,22]. Finally, there are models that propose to predict the degradation of the capacitance [18,20,23,24]. However, all of the above methods require experiment interruptions or can be sensitive to drastic changes in the ambient conditions. Most of the ongoing work focuses on the stability and material properties of different oxides to enhance the IS-FET performance but the operation of a FET device may be affected due to several process parameters that may make the results less reliable. Here, we present a detailed methodology to enhance the sensor reliability by aiming to get accurate values irrespective of the interacting oxide or FET operation.

In this paper, we propose a system for the absolute measurement of pH by simultaneously using two devices exposed to the same conditions and configured in a way that makes it possible to parameterize a synchronized response. We provide an analytical framework equivalent to an experimental mapping of the simultaneous current of two sensors that are used to describe the system and the methods that can determine the absolute pH irrespective of the experimental conditions. Our system improves the state-of-the-art by reducing the need of continuous calibrations and solving the problem of drift. The derivation of our analytical model is applied to protons, but it can be extended to correct the detection of any other ions that are selectively adsorbed and which suffer from the drift in the current.

The structure of this paper is as follows: the model to include the degradation region in the oxide capacitance is described in Section 2.1 followed by the equations to compute the depletion width. To describe experimental conditions, we applied the system to a high aspect ratio FinFET which enhances the sensitivity and reliability by a 3D gating while increasing the total surface area (Section 2.2). Section 3 outlines the main results and their discussion, including a meticulous analysis of the impact of degradation region on the depletion width and current as a function of the pH for ideal biosensors with two different oxides (Section 3.1) and non-ideal biosensors with SiO${}_{2}$ (Section 3.3). Finally, conclusions are drawn in Section 4.

## 2. Methodology

### 2.1. Oxide Capacitance

The migration of ions from the electrolyte into the dielectric is observed experimentally as a decrease of the capacitance resulting from the irreversible chemical transformation of a layer of the original material. The ions diffuse down to an effective depth that in some cases can be calculated for given experimental conditions [20,25]. The degraded material experiences a decrease of the dielectric constant which in some materials like Al${}_{2}$O${}_{3}$ reaches values of 20% of the original one [20]. The typical penetration depths of ions account for several nanometers in the span of hours depending on experimental conditions that include the pH of the electrolyte, the temperature and the ionic strength. In steady-state conditions, the degradation often leads to a fast transition and a complete failure of the device when leakage currents appear between the electrolyte and the semiconductor channel.

To simulate the degradation, both the effective dielectric constant and penetration depth can be adjusted phenomenologically to match the capacitance with several combinations that can provide a successful description of the sensor behaviour before the avalanche of leakage currents makes the device fail. In our model, we have modelled the degradation with a reduction to an arbitrary dielectric constant with a value 20% of the original one. The degraded region is associated with a corresponding effective penetration depth of the ions of *x* that is used to parameterize the degradation. To determine the absolute pH, we will consider two ISFET devices with different thickness *t*${}_{ox1}$ and *t*${}_{ox2}$ of the dielectric barrier, subjected to the same experimental conditions and thus, with the same penetration of the effective degradation *x* on both of devices.

Figure 1a shows sections of the interface between the electrolyte and the silicon channel using our model for two cases corresponding to device couples of SiO${}_{2}$ and HfO${}_{2}$, respectively. The dielectric barrier in each device can be considered to be made of two materials in series. The first material in contact with the electrolyte accounts for the degraded region with the adopted effective dielectric constant 20% of the original material and the total effective thickness *x* corresponding to the penetration of the degradation. The second material has the dielectric properties of the original material (SiO${}_{2}$ or HfO${}_{2}$) with a total thickness of ($t_{ox1}-x$) or ($t_{ox2}-x$) for the first and the second device or each sensor couple that will be used to determine the pH. Underlying the non-degraded dielectric in each device, there is the silicon channel as shown in gray in the schemes of Figure 1. Each sensor of absolute pH could consist of more than two of these devices which would be redundant but could help in improving precision. However, for simplicity in this work, we have considered only the use of two devices to determine the absolute pH. We also consider that all the devices with SiO${}_{2}$ or HfO${}_{2}$ dielectric barriers will have the same configurations (silicon dimensions, doping, length, etc.) except for the oxide thickness ($t_{ox1}$ = 5 nm and $t_{ox2}$ = 10 nm). For practical reasons, in the simulations of this work, we have considered a maximum penetration of ions degrading the oxide of 3 nm. We have also considered in our configuration a common reference electrode for both devices.

For each device, the total oxide capacitance ($C_{ox}$) is calculated as two capacitances in series including a capacitance without any degradation ($C_{ox}^{1}$) and another one with the degradation ($C_{ox}^{2}$):(1)$$\frac{1}{C_{ox}}=\frac{1}{C_{ox}^{1}}+\frac{1}{C_{ox}^{2}}=\frac{t_{ox}-x}{\varepsilon_{r}^{1}\varepsilon_{0}}+\frac{x}{\varepsilon_{r}^{2}\varepsilon_{0}}$$
where $\varepsilon_{r}^{1}$ is the relative dielectric constant of the original material, $\varepsilon_{r}^{2}$ is the relative dielectric constant of the degraded region material and $\varepsilon_{0}$ is the vacuum dielectric constant.

Accordingly, the total oxide capacitance for each sensor has been calculated from Equation (1):(2)$$C_{ox_{i}}=\frac{\varepsilon_{r}^{1}\varepsilon_{r}^{2}\varepsilon_{0}}{\varepsilon_{r}^{1}x+\varepsilon_{r}^{2}\left(t_{ox,1/2}-x\right)}$$
where the index *i* has been added to the total capacitance to determine $ox_{1}$ or $ox_{2}$ referring to the devices with the original dielectric of 5 nm or 10 nm, respectively.

Figure 1b shows the oxide capacitance calculated using Equation (2) as a function of the degraded region *x*, for the sensor interfaces with both thicknesses and materials. The observed behaviour corresponds to the decrease of the capacitance with increasing degradation depth *x*, which is equivalent to what observed in other works [25]. Regarding the total change in the oxide capacitance in the degradation range studied, it is much more pronounced for HfO${}_{2}$ than for SiO${}_{2}$. This is due to the higher dielectric constant $\varepsilon_{r}$ of HfO${}_{2}$ in comparison to the one of SiO${}_{2}$ (23.4 and 3.9, respectively [26]). Comparing the devices within the same material, as expected, the total variation of the capacitance is more pronounced for the configurations with 5 nm oxide thickness with respect to the thicker oxides of 10 nm, as the degraded region represents a larger part of the total dielectric thickness. Overall, it can be concluded that the HfO${}_{2}$ capacitance shows larger susceptibility to the degradation and the variations in the thickness due to the diffusion process when it is compared to the SiO${}_{2}$.

### 2.2. Calculation of the Field Effect in the Semiconductor Channel

To calculate the effect of the adsorbed charges on the device’s current, we consider the energy band diagram in the direction perpendicular to the surface of the oxide shown in Figure 1c. The model does not take into account the possible differences in the chemical potential between the semiconductor and the electrolyte; charges accumulated on the interface between the silicon and the dielectric barrier or phenomenon like the degradation of the reference electrode. When the semiconductor and the electrolyte are connected through a reference electrode and a gate voltage is applied between the two, it is possible to set the relation between the different potentials:(3)$$\Psi_{0}=V_{ox}-V_{G}+\Psi_{S}$$
where $\Psi_{0}$ is the oxide-electrolyte interface potential, $\Psi_{S}$ is the oxide-silicon interface potential, $V_{ox}$ is the potential drop across the oxide and $V_{G}$ is the external bias at the backgate.

The adsorption of protons in the oxide builds the potential $\Psi_{0}$, which is zero at the pH of point of zero charge ($pH_{pzc}$) where the adsorption and desorption processes are in equilibrium and which can be calculated from their proton affinities $pK_{a}$ and $pK_{b}$ ($pH_{pzc}=\frac{pK_{a}+pK_{b}}{2}$). This potential has a Nerstian response which has been well described in literature [6]:(4)$$\frac{\partial \Psi_{0}}{\partial pH}=-2.303\frac{k_{B}T}{q}\alpha$$
where $k_{B}$, *T*, and *q* are the Boltzmann constant, the temperature, and the electron charge, respectively. The sensitivity parameter α depends on the chemical buffering capacity of the dielectric surface in contact with the electrolyte and the response of the ions in solution that will create the double layer capacitance. Often α is considered only in the linear range of the sensor, and in the ideal case it can be approximated by 1, showing the theoretical maximum variation (slope) of the surface potential with respect to the pH ($\frac{\partial \Psi_{0}}{\partial pH}=-2.3\frac{k_{B}T}{q}$).

To consider the chemical response of the biosensor (non-ideal sensor), α can be calculated making use of an iterative method with $\Psi_{0}$ making use of the dissociation model and Gouy-Chapman-Stern theory [6,7]. This more realistic assumption has been taken into account in our analytic model in Section 3.3. In this method, the sensitive parameter is calculated making use of the diffusion capacitance ($C_{diff}$), which depends on the electrolyte properties (considering the double layer and the Stern capacitances), and the intrinsic buffering capacitance of the dielectric ($\beta_{diff}$), which depends on the number of binding sites on the sensing surface $N_{s}$ and their corresponding proton affinities $pK_{a}$ and $pK_{b}$:(5)$$\alpha =\frac{1}{1+\frac{2.303k_{B}TC_{diff}}{q^{2}\beta_{diff}}}$$

$\Psi_{0}$ in Equation (4) changes relative to the pH at $pH_{pzc}$, i.e. the acidity in the electrolyte at which the equilibrium of protonated and deprotonated species in the surface accounts for neutrality. The conduction band-bending at this $pH_{pzc}$ depends only on the chemical equilibrium at the interface between the dielectric and the semiconductor interface, which is accounted by the flat band potential ($V_{fb}$). It is always possible normalization of the device current at $pH_{pzc}$ to account for the band bending due to $V_{fb}$. At $pH\neq pH_{pzc}$, $\Psi_{0}$ will be equilibrated in the semiconductor channel by the mobile charge carriers that will migrate and generate a potential within the semiconductor ($\Psi_{S}$).

The term $V_{ox}$ in Equation (3) accounts for the energy accumulated across the dielectric barrier. It can be expressed using the charge in the semiconductor side and considering a planar condenser:(6)$$V_{ox}=\frac{qN_{A}W_{D}}{C_{ox}}$$
where $C_{ox}$ is the area capacitance of the dielectric barrier (typically a metal oxide) described in Section 2.1, *q* is the elementary charge, $N_{A}$ is the density of dopants in the semiconductor and $W_{D}$ is the region in the semiconductor channel depleted from carriers shown in Figure 1c in darker grey color. $W_{D}$ can be derived solving the Poisson equation for $\Psi_{S}$ with a planar configuration:(7)$$\frac{\partial^{2}\Psi_{S}}{\partial x^{2}}={\left.\frac{-qN_{A}}{\varepsilon_{s}\varepsilon_{0}}\right|}_{x=0}^{x=W_{D}}⟶\Psi_{S}=\frac{qN_{A}W_{D}^{2}}{2\varepsilon_{s}\varepsilon_{0}}$$
where $\varepsilon_{s}$ is relative dielectric constant of the semiconductor. We have replaced $\varepsilon_{s}$$\varepsilon_{0}$=$\varepsilon_{Si}$ as p-type doped Silicon is used as a semiconductor channel for this work. Note that $W_{D}$ changes the region populated with carriers and thus can modulate the conductivity of the FET channel. Combining Equations (3), (6) and (7), we have the following dependence:(8)$$\Psi_{0}+V_{G}=\frac{qN_{A}W_{D}}{C_{ox}}+\frac{qN_{A}W_{D}^{2}}{2\varepsilon_{Si}}$$

In Equation (8), $W_{D}$ changes with respect to the pH through the dependence of $\Psi_{0}$ with the acidity expressed in Equation (4) and so it is possible to get a final expression for $W_{D}$ as a function of the pH:(9)$$W_{D}=-\frac{\varepsilon_{Si}}{C_{ox}}+\sqrt{{\left(\frac{\varepsilon_{Si}}{C_{ox}}\right)}^{2}+2\left(\frac{\varepsilon_{Si}}{qN_{A}}\right)\left(\Psi_{0}+V_{G}\right)}$$

A drift in the current will be observed due to the dependence of $W_{D}$ with the degradation of the different parameters. In particular, the parameters from Equation (9) which can be responsible for the drift are: (i) the changes in the dielectric material and thus in $C_{ox}$ due to the possible penetration of ions or modifications of the dielectric (Section 2.1); and (ii) the changes in the sensitivity (α) of the material, mainly due to the modifications in $\beta_{diff}$ because of the degradation of the surface with absorbed molecules that change the number of sites ($N_{S}$) for the binding of protons. In this work, the sensitivity has been calculated assuming the ideal sensor (in order to equally compare the oxides) and making use of an iterative method with respect to $\Psi_{0}$ to consider the real sensor. Accordingly, we have focused on studying the impact of different penetration of ions in $C_{ox}$.

## 3. Results and Discussion

### 3.1. Impact of the Dielectric Degradation on the Depletion Width of Different Materials

Based on the analytical model described above, we have simulated the four devices detailed in Section 2.1 grouped in couples having two thicknesses (5 nm and 10 nm) for each material (SiO${}_{2}$ or HfO${}_{2}$). We have calculated the effect of the degradation in *W*${}_{D}$ as a function of the pH. As a first step, to simplify the study of the drift from other effects like the combination of the chemical affinity with the changes in the electrolyte, we have considered the case with ideal sensitivity (α=1).

We used Equation (9) to calculate the parameters of the semiconductor channel, considering a desirable dynamic range from 2 to 12. Thus, considering a p-doped semiconductor channel that is going to be depleted in acidic conditions, we calculated a bias external voltage $V_{G}$ necessary to have full conductivity ($W_{D}=0$ nm) at pH = 12, and calculated the value for both oxide materials. $N_{A}$ was chosen to have a depletion region of $W_{D}=100$ nm at pH = $pH_{pzc}$ considering the devices with $t_{ox1}$ = 5 nm.

Figure 2 shows $W_{D}$ vs. pH for the interfaces described in Figure 1 using the designated colour codes. A tone scale convention from darker to a lighter colour for increasing *x* has been added and will be maintained hereafter. As expected, $W_{D}$ decreases with pH in all the devices as a result of the effect of the adsorbed protons. Comparing SiO${}_{2}$ and HfO${}_{2}$, the latter has a larger variation across the pH dynamic range due to the higher dielectric constant. The impact of the drift caused by the degradation on the pH determination by each of the devices is clearly observed in these graphics, as for a single depletion width, there are a broad number of possible pH values corresponding to different states of degradation in the material. For instance, if the constant *W*${}_{D}$ = 60 nm is considered (solid orange line in Figure 2), the pH uncertainty between the cases of no degradation (*x* = 0 nm) and the maximum degradation considered (*x* = 3 nm) are ΔpH = 3.15 and ΔpH = 3.30 for the SiO${}_{2}$ dielectric with $t_{ox1}$ = 5 nm and $t_{ox2}$ = 10 nm, respectively; whereas the uncertainty is dramatically reduced to ΔpH = 0.30 and ΔpH = 0.31 for the HfO${}_{2}$ dielectric with $t_{ox1}$ = 5 nm and $t_{ox2}$ = 10 nm, respectively. In both cases, ΔpH is slightly higher for the device with $t_{ox1}$ = 10 nm.

On the other side, as considering Figure 1b, HfO${}_{2}$ devices can offer a better pH resolution as the current range of pH values possible relative to the total current variation in the dynamic range, is much more restricted than for SiO${}_{2}$. In addition, even if it is not taken into account by these simulations, the chemical stability of HfO${}_{2}$ largely exceeds the one of SiO${}_{2}$, and thus is less prone to ion penetration which makes that the degradation occurs in longer time periods. Elseways, SiO${}_{2}$ has proportionally a larger variation of $W_{D}$ as the degradation increases. In order to resolve the absolute pH, we intend to determine the degradation considering the current from a dual device composed of the two sensors with one of the two materials that we had calculated. In this sense, SiO${}_{2}$ may have the advantage to determine an absolute pH as it provides proportionally larger current contrasts within a given pH range.

### 3.2. Determination of Absolute pH from Current Acquisition in FET Sensors

To illustrate the determination of pH in a case scenario, we used our model to calculate the response of a pair of sensors with the geometry of a high aspect ratio FinFET shown in Figure 3a. This ISFET geometry has been recently proposed by us as a robust and advanced design for a biosensor [27]. Similarly to single Silicon-Nanowire (SiNW), this geometry offers a three-dimensional direct gating which is advantageous with respect to typical planar devices or extended gates. Respect to the nanowires (NWs), the high aspect ratio FinFETs can also improve: (i) the reproducibility of the sensitivity for ion sensing (pH), (ii) the total signal, and (iii) the linearity of the current response. Moreover, high aspect ratio FinFETs have better linearity and a smaller footprint if compared to NW arrays. Due to the planar configuration of the conduction channel, the influence of small defects in pH sensing is localised and negligible for the sensor signal if compared to their influence in nanoscale SiNWs. For our work, we have chosen the device dimensions similar to the one shown in Figure 3a, where the width *W* was 200 nm, the height *h* was 2 μm and the length L was 10 μm.

For a given FinFET, $W_{D}$ can be related to the measured current depending on the geometry of the sensor considering that the size of the channel is diminished across the cross-sectional area by $W_{D}$ in all the directions perpendicular to the surfaces in contact with the electrolyte, and then the total current ($I_{SD}$) can be calculated as:(10)$$I_{SD}=\sigma \frac{A}{L}V_{SD}$$
where σ, *A*, and *L* are the conductivity (material property), the cross-section, and the length of the silicon channel, respectively. At the point of zero charge, *A* coincides with the geometrical dimensions of the FinFET channel (A=W×h) as pH increases [$A=\left(W-2W_{D}\right)\times \left(h-W_{D}\right)$]. In this way, $W_{D}$ is connected to the experimental data using the original geometrical cross-section and the actual resistance of the channel (ρ=1/σ). Given the large aspect ratio, we have considered $h>>W_{D}$ and thus we have approximated as $A=\left(W-2W_{D}\right)\times \left(h\right)$. This possibility to neglect the depletion width in one direction, is indeed the origin of the higher linearity of the high aspect ratio FinFET respect to NWs described in our works [27,28]. Figure 3a shows two SEM pictures from different perspectives of a typical high aspect ratio FinFETs in which we have included schematics showing the electrical connections and the geometrical parameters *W*, *h* and *L*.

Figure 3b shows $I_{SD}$ vs. pH for the pair of devices with silica dielectric at three different degradation points (*x* = 0.5 nm, *x* = 1.5 nm and *x* = 2.5 nm shown in darker to lighter colours and using solid blue lines for the thinner sensor and green dashed lines for the thicker sensor). We illustrate that at an arbitrary pH, the acidity can be unequivocally determined using the current values that intersect for example the orange lines in Figure 3b–d that mark a constant pH for values 3 and 10. For each of these pH values and for each state of degradation (*x*), Figure 3b provides a pair of current values that will be observed on the pair of sensors at the intersection with the indicated orange lines with each of the curves respective to each degradation *x*. For each sensor alone, there are several combinations of pH vs. degradation *x* that provide such currents values, but only at one point, a pair of currents converge with the same pH and degradation *x*. Figure 3d shows the equivalent situation for pH 10.

Both Figure 2 and Figure 3 show that at more basic pH values, the differences in signal between devices of the same material becomes smaller, and thus discriminating the value of the currents for each state of degradation becomes more difficult depending on the values of the noise signal. By this effect, the determination of pH is also more affected by the noise signal at a more basic pH as the acidity has not acted on the surface potential that builds the depletion width $W_{D}$. This becomes also apparent comparing the range of pH variation for a given current in Figure 3c,d. The range of pH in the degradation span of our studies for each current is nearly three times larger for the pair of current values acquired at pH 3 (Figure 3c) than for the ones at pH 10 (Figure 3d), showing that the degradation can depend less in the measuring error in the first case. It is to be noted, that the current map calculations shown in Figure 3 using our simultaneous of current vs. pH, are equivalent to an experimental-mapping in a pair of devices with the same fabrication parameters except for oxide thickness, and where the simultaneous current response would be mapped during the degradation of the oxide. In such a case, we would bet a current map equivalent to Figure 3b. Given the broader response of SiO${}_{2}$ to the degradation, it would require less precision on the determination of the current to obtain a match in the current response to a single pH compared to materials with less change with degradation, as for example the case of HfO${}_{2}$. However, the lifetime and the variability of the sensor over time would still be beneficial for the material with higher chemical stability and dielectric constant.

### 3.3. Implementation of the Proton Affinity on the Sensor Response for Non-Linear Sensitivities

The simplification of ideal sensitivity α=1 predicts an excessive sensitivity and fails to describe the effect of saturation of proton adsorption that occurs at acidic concentrations. Materials like $SiO_{2}$ decrease in α due to the saturation of the silanol groups accepting protons. In this section, we are presenting results calculating the chemical response of the sensor to describe the best way to operate these devices. To include the chemical sensitivity of the materials and the effects of the electrolyte, it is necessary to include the values of $\beta_{diff}$ and $C_{diff}$. The buffering capacitance is calculated from the site-binding model:(11)$$\beta_{diff}=2.303\cdot a_{H_{s}}\cdot N_{S}\frac{K_{b}a_{H_{s}}^{2}+4K_{a}K_{b}a_{H_{s}}+K_{a}K_{b}^{2}}{{\left(K_{a}K_{b}+K_{b}a_{H_{s}}+a_{H_{s}}^{2}\right)}^{2}}$$
where $a_{H_{s}}$ is the surface proton activity that depends on the bulk pH and on the surface potential. $C_{diff}$ can be estimated with the Gouy-Chapman-Stern approximation. It is calculated as the Stern capacitance (C${}_{Stern}$) in series with the double layer capacitance (C${}_{DL}$). C${}_{Stern}$ = 0.8$\frac{F}{m^{2}}$ has been considered in this work in order to consider a realistic behaviour of an ISFET [6]. C${}_{DL}$ is calculated by:(12)$$C_{DL}=Q_{o}cosh\left(\frac{q\Psi_{0}}{2k_{B}T}\right)=\sqrt{\frac{2\varepsilon_{W}I_{0}N_{avo}q^{2}}{k_{b}T}}cosh\left(\frac{q\Psi_{0}}{2k_{B}T}\right)$$
where $\varepsilon_{W}$ is the electrolyte permittivity, I${}_{0}$ is the ion concentration in mol/L, and N${}_{avo}$ is the Avogadro constant. As $\beta_{diff}$ and $C_{diff}$ depend on the surface potential α and $\Psi_{0}$ are computed in a self-consistent loop described above.

Figure 4 shows the relationship between α, $\Psi_{0}$ and pH. The model is able to reproduce the saturation of the surface potential observed in our previous experiments [27,28,29]. At pH = 2, which is the point of zero charge for SiO${}_{2}$, α has its minimum value. The origin of the surface potential $\Psi_{0}$ is also set equal to 0 at pH 2. Figure 4 shows that with increasing of the pH, the value of α becomes close to 1.0 (α must have value between 0 and 1) and the surface potential $\Psi_{0}$ increases to a higher negative value. As the acidity is increased, the decrease in α results in the saturation of change in $\Psi_{0}$. It is also to be noted, that contrary to what is assumed in most cases, the behaviour of $\Psi_{0}$ is not linear through the pH range, and that has singularities due to the interplay of proton affinities with the double-layer capacitance.

Figure 5 shows $I_{SD}$ vs. pH for a device couple of SiO${}_{2}$ with corrected α. As in the case of the ideal sensitivity, the current values in both devices converge to the maximum at basic conditions due to the vanishing $W_{D}$. Contrary, when the pH is very acidic (pH = 2 or 3), there is a larger drift of the current values with *x*. The effect of the drift is even larger for the device with $t_{ox1}$ in comparison to the $t_{ox2}$. This is expected due to the larger proportion of degraded material in the device with oxide thickness equals to 5 nm. Another interesting point is that both devices have almost identical current profile for all pH values at maximum degradation of 3 nm ($I_{SD}$ vs. pH curves with lighter colours). Hence, it seems that once the degraded region of the oxide dominates the contribution of the capacitance. The effects of the saturation of the sensitivity α are also observed in the acidic range for both currents simulated in Figure 5 as the variation of $I_{SD}$ vs. pH decreases as the pH becomes more acidic. This loss of sensitivity affects also the determination of pH, as for a given noise signal, more pH values will fall within the range of error. However, this is a property of the material observed in the saturation of the surface potential in Figure 4 which cannot be resolved with a different operation mode.

The lines of constant $I_{SD}$ at 20% of the total conductance ($I_{SD}$ = 0.2 μA) are indicated as horizontal orange lines in Figure 5. It can be noticed, that the pH uncertainty associated to that measurement is much greater for the device with thinner oxide.

### 3.4. Optimisation of pH Determination Using a $I_{SD}$ Follower in One of the Sensors

The current response obtained in Figure 5a,b can be used to reproduce the plan of action described at the end of Section 3.2 to obtain the absolute pH. However, using a constant gate voltage is detrimental to the accuracy at more basic pH values due to the similar values between currents at different *x* because of the small values of $W_{D}$. The traditional method to measure the acidity follows the surface potential $\Psi_{0}$ by compensating with a voltage bias applied between the channel and the reference electrode to maintain a constant current, usually closed to one obtained with the threshold voltage of the transistor (maximum *W*${}_{D}$), but not too low as to increase the signal to noise ratio.

Figure 6a,b show the calculation of the gate voltage correction to maintain a current of 0.2 μA (equivalent to a *W*${}_{D}$ of 80 % of the width of the sensor) as a function of pH and for all the states of degradation within the range of our study for the devices with thinner and thicker dielectrics, respectively. The different curves of $V_{G}$ vs. pH for each state of degradation are parallel to each other, showing an opposite behaviour to $\Psi_{0}$ (shown in Figure 4) to compensate the charge accumulated due to the pH.

In order to use Figure 6a,b as a map of values to determine the pH, we have to take into consideration that there is only a common reference electrode in the system. Consequently, only one of the devices can be kept at a constant current $I_{DS}$. Here, we have arbitrarily chosen to maintain constant the device with the smaller oxide thickness, and use the map in Figure 6a corresponding to a particular pH and state of degradation, while using the obtained values of $V_{G}$ and parameter of degradation *x* to calculate the current that corresponds in the second device. Figure 6c,d show the possible values of pH vs. degradation that could be obtained at the values mapped for pH 3 and 10, respectively, for both sensors shown in blue and green for the 5 nm and 10 nm sensors, respectively. We have extracted the values of $V_{G}$ and current of the second sensor obtained at the levels of degradation of 0.5 nm, 1.5 nm and 2.5 nm. It can be observed that equally to the method of the current, for each pair of devices, there is a single point that determines the pH and the parameter of degradation. Comparing the slopes obtained in the pH determination using the current output Figure 3c,d with the ones obtained with the mapping of $V_{G}$ and the current of the second sensor in Figure 6c,d, we can notice that the later have a steeper slope. This signals also the better determination as determine range of degradation corresponds to a shorter range of pH with the best precision acquired in Figure 6c of the determination only with the current. Thus, using current as the mapping parameter for an unknown variable “oxide degradation” and controlled variable “operating bias”, we were able to accurately determine the pH value with the help of two similar devices with different oxide thickness.

## 4. Conclusions

In this paper, we have developed an analytical model to calculate the surface potential and the current response of ISFETs to pH. We have implemented the effect of the degradation at the dielectric barrier that induces the current drift. The derived model used a capacitance representing the degraded region which is adjusted with a phenomenological effective dielectric constant and depth connected in series with the capacitance of the rest of non-degraded material with the original properties. We calculated the response of the degradation of the capacitance for two materials, SiO${}_{2}$ and HfO${}_{2}$ as examples of low and high dielectric constants, respectively. The relative effect of the degradation is higher for materials with lower dielectric strength. Further, without any correction, the materials with a higher dielectric constant have less uncertainty of the measured pH.

Using the modification of the capacitance with degradation, we propose a method to determine the absolute pH using a mapping of dual sensor response. In our paper, we have used a mapping with calculations equivalent to a mapping that would be produced experimentally with reproducible devices. To simulate the effects of the chemical response of the materials, we have implemented the site-binding model interacting with stern and double layer capacitances. This model does not take into account the modification of binding sites at the interface of the dielectric and the electrolyte. We have shown that using a common reference electrode at constant voltage, the current values are less accurate to determine the pH at basic pH where there is less action of the acid and less depleted region in the semiconductor. This effect can be partially corrected using the voltage of the reference electrode as a current follower for one of the devices. However, in the case of materials like SiO${}_{2}$, the effect of site-binding saturation at acidic pH also causes a decrease in the sensitivity, which affects also the possibility to determine the absolute pH.

In summary, we have shown a method to determine the absolute pH using dual measurements from two sensors, which can be a breakthrough for the applications of ISFET in physiological monitoring, or quantification of ion activity.

## Figures and Tables

**Figure 1 sensors-21-05190-f001:**
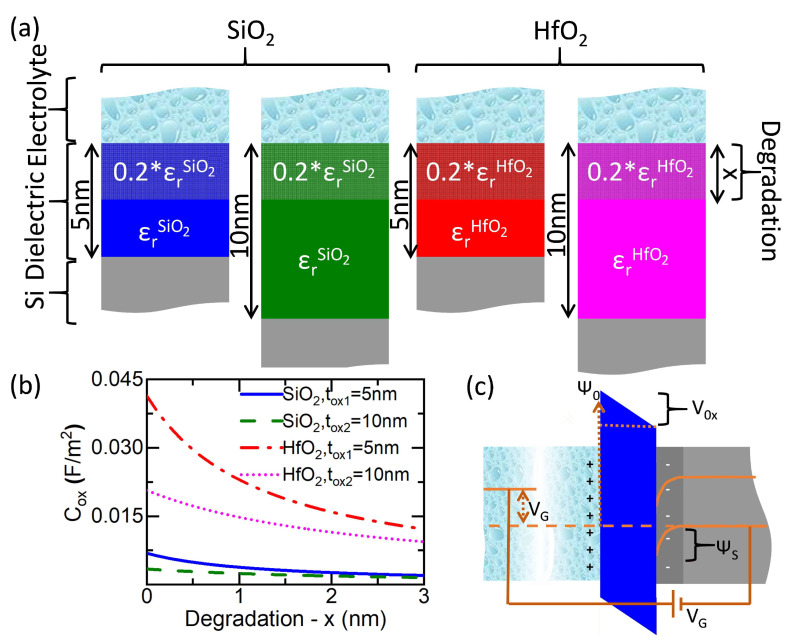
(**a**) Schematic representation that shows how the oxide capacitance is modelled considering a degraded region. Two different couple of devices are described in the diagrams including the combinations for two different oxide materials in the dielectric region. For each device couple, we have considered two oxide thicknesses: (blue) $t_{ox1}$ = 5 nm and SiO${}_{2}$, (green) $t_{ox2}$ = 10 nm and SiO${}_{2}$, (red) $t_{ox1}$ = 5 nm and HfO${}_{2}$, and (magenta) $t_{ox2}$ = 10 nm and HfO${}_{2}$. This color notation to identify each device has been kept the same throughout the whole paper. (**b**) Total capacitance vs. degradation using the penetration depth of the degrading charges as a parameter of the degradation. (**c**) Schematic of the energy band alignment along one interface in a generic ISFET sensor.

**Figure 2 sensors-21-05190-f002:**
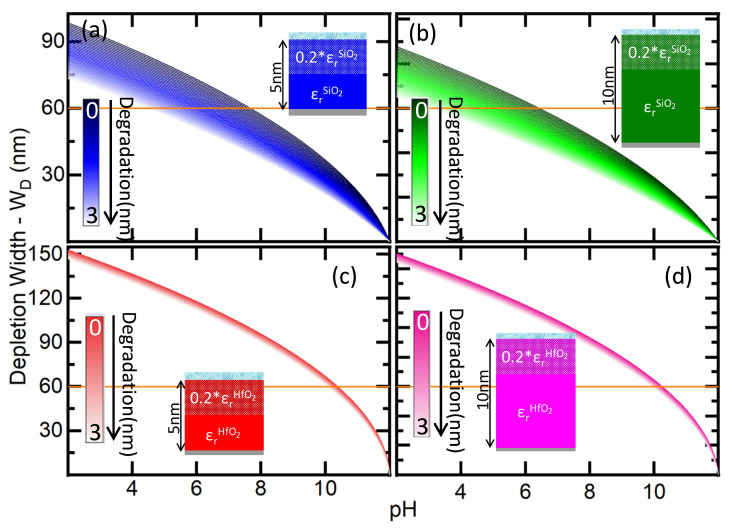
Depletion Width ($W_{D}$) as a function of the pH considering different degraded region in the oxide (*x*) from *x* = 0 nm (non-degraded oxide) to *x* = 3 nm for the two different oxides [(**a**)/(**b**) SiO${}_{2}$ and (**c**)/(**d**) HfO${}_{2}$] and two different ideal biosensors (α = 1) which main difference is the oxide thickness [(**a**)/(**c**) $t_{ox1}$ = 5 nm and (**b**)/(**d**) $t_{ox2}$ = 10 nm]. The solid orange line represents the example of the variation of the pH range for a constant *W*${}_{D}$ = 60 nm. The calculated gate bias for the above simulations is 0.3825 V.

**Figure 3 sensors-21-05190-f003:**
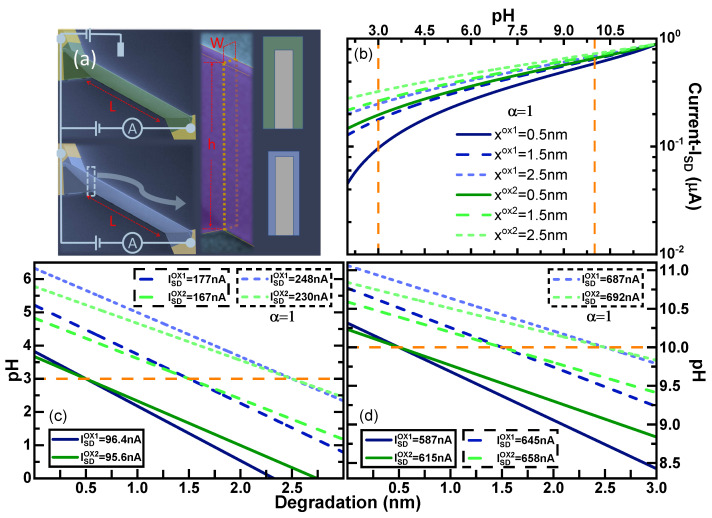
(**a**) SEM pictures from a typical FinFET device fabricated in LIST, schematically showing the electrical connections and the dimensions. In our work W, h and L have been chosen 200 nm, 2 μm and 10 μm, respectively. (**b**) Current ($I_{SD}$) as a function of the pH considering three degraded regions in the oxide (*x* = 0.5 nm, *x* = 1.5 nm, and *x* = 2.5 nm) for SiO${}_{2}$ and two different ideal biosensors (α = 1) having main difference is the oxide thickness ($t_{ox1}$ = 5 nm and $t_{ox2}$ = 10 nm). (**c**,**d**) Calculated pH as a function of the degradation (*x*) for SiO${}_{2}$ and two different ideal biosensors (α = 1) having main difference is the oxide thickness ($t_{ox1}$ = 5 nm and $t_{ox2}$ = 10 nm). The pH has been calculated considering the *W*${}_{D}$ given by Equation (9) with three degraded regions in the oxide (*x* = 0.5 nm, *x* = 1.5 nm, and *x* = 2.5 nm) and an initial (**c**) pH=3 and (**d**) pH = 10. The solid orange line represents the constant initial pH. Drain bias equals 50 mV and calculated gate bias equals 0.5914 V are used for the simulation.

**Figure 4 sensors-21-05190-f004:**
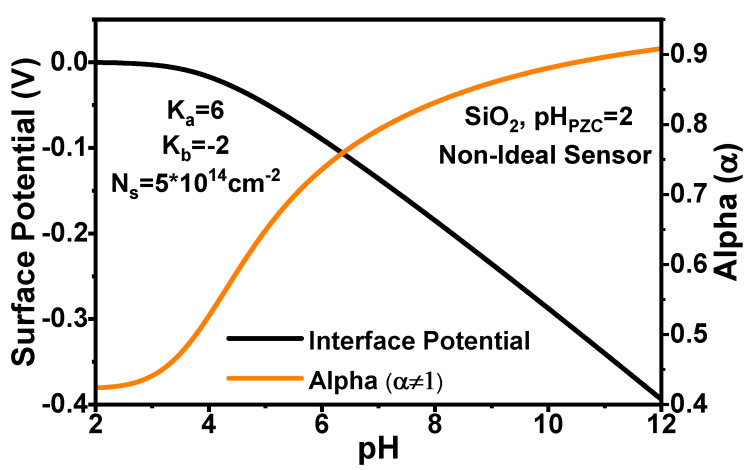
Surface potential ($\Psi_{0}$) and sensitivity parameter (α) as a function of the pH calculated using the iterative method for a non-ideal sensor considering only SiO${}_{2}$.

**Figure 5 sensors-21-05190-f005:**
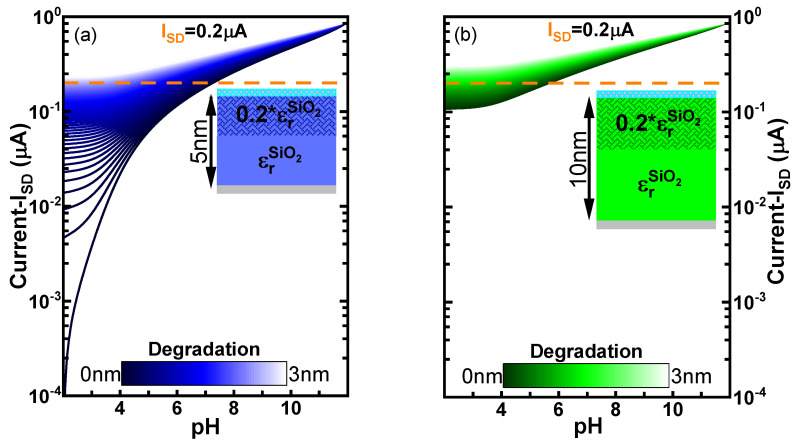
Current ($I_{SD}$) as a function of the pH considering different degraded region in the oxide (*x*) from *x* = 0 nm (non-degraded oxide) to *x* = 3 nm for SiO${}_{2}$ and two different non-ideal biosensors (α is self-consistently computed using the iterative method as shown in Figure 4) which main difference is the oxide thickness ((**a**) $t_{ox1}$ = 5 nm and (**b**) $t_{ox2}$ = 10 nm). The solid orange line represents the example of the variation of the pH range for a constant *I*${}_{SD}$ = 0.2 μA. Drain bias equals 50 mV and calculated gate bias equals 0.3825 V are used for the simulation.

**Figure 6 sensors-21-05190-f006:**
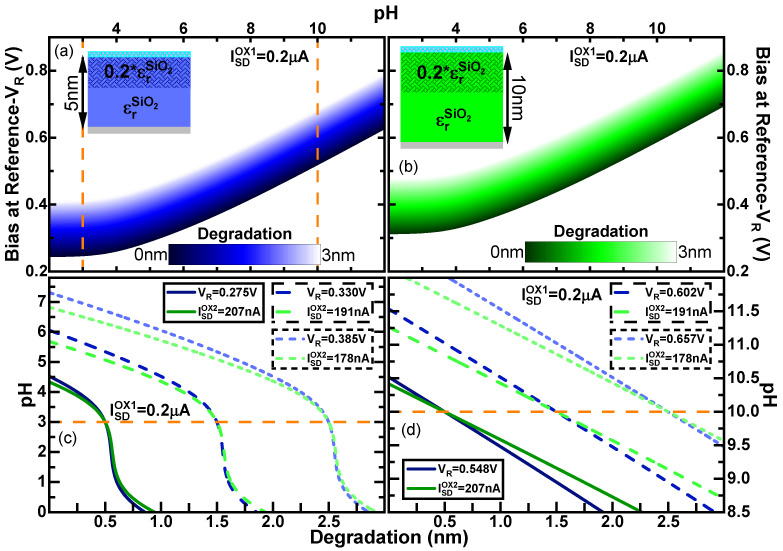
(**a**,**b**) External bias at the backgate (*V*${}_{G}$) as a function of the pH calculated using the Equation (3) for a constant *I*${}_{SD}$ = 0.2 μA considering different degraded region in the oxide (*x*) from *x* = 0 nm (non-degraded oxide) to *x* = 3 nm for SiO${}_{2}$ and two different non-ideal biosensors which main difference is the oxide thickness ((**a**) $t_{ox1}$ = 5 nm and (**b**) $t_{ox2}$ = 10 nm). (**c**,**d**) Calculated pH value as a function of the oxide degradation (*x*) for SiO${}_{2}$ [having different external bias (*V*${}_{G}$) and current (I${}_{SD}$)] and two different non-ideal biosensors which main difference is the oxide thickness ($t_{ox1}$ = 5 nm and $t_{ox2}$ = 10 nm). The *V*${}_{G}$ has been calculated using Equation (3) considering the *W*${}_{D}$ given by Equation (9) with three degraded regions in the oxide (*x* = 0.5 nm, *x* = 1.5 nm, and *x* = 2.5 nm) and an initial (**c**) pH = 3 and (**d**) pH = 10. The solid orange line represents the constant *V*${}_{G}$ in which the curves for both devices cross. In all the figures (**a**–**d**), α is self-consistently computed using the iterative method as shown in Figure 4.

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
