# Peer review of "Comprehensive Analytical Modelling of an Absolute pH Sensor"

_sensors, 2021, doi:10.3390/s21155190_

Round 1

Reviewer 1 Report

  • A list of abbreviation is to be added
  • In some position, there is a space between the number and unit (for example, in line 102: 5 nm and line 104: 3 nm) while in others no space (line 197: 5nm and line 207: WD = 100nm). So, for consistency use one style.
  • tox,1 or tox1 (For example, in line 102 and caption of Figure 1 it is given as tox,1 while in line 94 it is written as tox1)?? please use one style in writing it. Same applies for tox2 and tox,2

Line 31  delete the ? shown after ref 13; 13?-15

Figure 1 (a)         Check the chemical formula of SiO2 and HfO2 in the diagram of the dielectric region; the 2 to be as subscript in the formula.

Line 109 -110 and line 121           The symbol of relative/vacuum dielectric constant is not written as it is shown in equation (1). In the equation it is written as ε while in the text as ϵ.

Line 113               Equation 1 is supposed to be Equation 2

Line 138-139      Ѱs  is the ……. (it is given in equation 3 but not mentioned in the text what it stands for)

Line 141               …where the adsorption and desorption are in equilibrium ..

Line 169               The term Vox in Equation 4 …! Do you mean Equation 3?

Equation (8)        εSi was shown in the combined equation without giving any explanation in the text about it or why it replaces εSε0

238        SiNWs is mentioned here for the first time, so the full name of this abbreviation is to be given.

Line 240               ..Respect to the nanowires (NWs), …

Line 262               …,and then the total current (ICD) can be …

Line 285               …than for ones at pH 10 (3(d)),…

Line 287               …shown in figure 3 using…

Equation 11        Ka or ka?? Kb or kb?? Check and edit accordingly!

Line 299              As βdiff and …

Line 303               Delete the duplicate word ‘’experiments’’.

Line 318-319      Check this sentence?? Seems to be not complete or missing some words.

Line 335 and Line 344       Potential φ0 or Ѱ0?? Check it!

Line 351               Figure 6(c) and (d) show …

Line 388               ..causes a decrease in the sensitivity, which affects…

References         

Some journal names are shown as abbreviations while others written with full name. Check and edit by using one style.

Author Response

  • A list of abbreviation is to be added

Comment: Please find below the required List of Abbreviations, however we did not know the way the editors would like to include it, or if it is necessary as they are also defined within the text:

ISFETs- Ion Sensitive Field-Effect Transistors

MOS- Metal-Oxide-Semiconductor

DNA- Deoxyribonucleic Acid

FinFET- Fin Field-Effect Transistors

Cox- Oxide Capacitance

Vox- Voltage across Oxide

Vg- Gate Voltage

pHpzc- pH at Point-of-Zero-Charge

Cdiff- Diffusion Capacitance

βdiff- Intrinsic Buffer Capacitance

Vfb- Flat-Band Voltage

WD- Depletion Width

Na/Ns- Surface States

tox- Oxide Thickness

SiNW- Silicon-Nanowire

ISD- Source-Drain Current

CDL- Double Layer Capacitance

  • In some position, there is a space between the number and unit (for example, in line 102: 5 nm and line 104: 3 nm) while in others no space (line 197: 5nm and line 207: WD = 100nm). So, for consistency use one style.
  • tox,1or tox1 (For example, in line 102 and caption of Figure 1 it is given as tox,1 while in line 94 it is written as tox1)?? please use one style in writing it. Same applies for tox2 and tox,2 Line 31 delete the ? shown after ref 13; 13?-15
  • Figure 1 (a) Check the chemical formula of SiO2 and HfO2 in the diagram of the dielectric region; the 2 to be as subscript in the formula.
  • Line 109 -110 and line 121 The symbol of relative/vacuum dielectric constant is not written as it is shown in equation (1). In the equation it is written as εwhile in the text as Ïµ.
  • Line 113 Equation 1is supposed to be Equation 2
  • Line 138-139 Ѱsis the ……. (it is given in equation 3 but not mentioned in the text what it stands for)
  • Line 141 …where the adsorption and desorption arein equilibrium ..
  • Line 169 The term Vox in Equation …! Do you mean Equation 3?
  • Equation (8) εSi was shown in the combined equation without giving any explanation in the text about it or why it replaces εSε0
  • 238 SiNWsis mentioned here for the first time, so the full name of this abbreviation is to be given.
  • Line 240 ..Respect to the nanowires (NWs), …
  • Line 262 …,and then the total current (ICD) can be …
  • Line 285 …than for ones at pH 10 (3(d)),…
  • Line 287 …shown in figure3 using…
  • Equation 11 Ka or ka?? Kb or kb?? Check and edit accordingly!
  • Line 299 As βdiffand …
  • Line 303 Delete the duplicate word ‘’experiments’’.
  • Line 318-319 Check this sentence?? Seems to be not complete or missing some words.
  • Line 335 and Line 344 Potential φ0or Ñ°0?? Check it!
  • Line 351 Figure 6(c) and (d) show …
  • Line 388 ..causes a decrease in the sensitivity, which affects…
  • References: Some journal names are shown as abbreviations while others written with full name. Check and edit by using one style.

Comment: We are particularly grateful for the degree of detail given to our manuscript and for these minor concerns. All of them have been accounted for in the new version.

Reviewer 2 Report

No comment!

Author Response

We thank the reviewers for the suggestions that aim to improve the clarity and quality of the manuscript. We have corrected the manuscript as per the suggestions. We have reconfigured and reframed wherever possible for the betterment of the manuscript.

Reviewer 3 Report

The study is very well documented and presented, the importance of the subject and its novelty are highlighted. The mathematical apparatus used for the simulation and the presented results are very well explained by the authors. The figures are necessary, sufficient, and properly explained. The quality of all the figures included in this manuscript is good.

There are also some typos such as pasted or missing letters as well as small English mistakes. Thus, I recommend that the authors carefully review the manuscript.

In conclusion, the manuscript sent by César Pascual García and coworkers can be published in Sensors after a minor revision.

Author Response

(The authors gave the same response as above.)

Reviewer 4 Report

This manuscript described the improvement of the accuracy of pH measurement of general field-effect chemical sensors. The authors focused on the degradation of the sensor surface, its influence, and also the way to compensate for it. The reviewer agrees with the work is unique and important to the development of chemical sensors, however, there are several points to be considered for the publication.

Major:

  • In section 2.2, it seems to be that the type of the semiconductor is not defined. In addition, the band diagram in Fig. 1c is wrong in several points. The authors should check the consistency of signs/definitions of variables in 2.2, again.
  • In section 2.3, the description of the FinFET would be redundant. It unfocuses the importance of the work. And also, the border of the numerical calculation and the experimental result is unclear. If any experimentals were performed, the details of that (including how the results were combined with calculations) should be described.
  • The reviewer recommend to think about the phrase of “absolute pH value”. Generally, the measurement of field-effect chemical sensors detect “pH changes”.

Minor:

  • The abstract would better explain “the paper is about IS-FET” (otherwise, more widely field-effect chemical sensors) for the readers.
  • The comparison of the dielectric constants and chemical stability of general materials for the gate passivation layer (such as Silicon nitride, Tantarum oxide, Aluminum oxide) with HfO2 would emphasize the importance
  • The caption of Fig.6c – d begins as “Calculated external bias … as a function of the degradation…”, but the plot is actually converted in pH scale. It is not kind for readers.
  • The word of “mapping” is unclear.

(over)

Author Response

Comments and Suggestions for Authors

This manuscript described the improvement of the accuracy of pH measurement of general field-effect chemical sensors. The authors focused on the degradation of the sensor surface, its influence, and also the way to compensate for it. The reviewer agrees with the work is unique and important to the development of chemical sensors, however, there are several points to be considered for the publication.

We thank the reviewer for the suggestions that aim to improve the clarity and quality of the manuscript. We have corrected the manuscript as per the suggestions. We have reconfigured and reframed wherever possible for the betterment of the manuscript.

Major:

  • In section 2.2, it seems to be that the type of the semiconductor is not defined. In addition, the band diagram in Fig. 1c is wrong in several points. The authors should check the consistency of signs/definitions of variables in 2.2, again.

Comment: The type of semiconductor is p-type doped (1017cm-3) Silicon. The correction has been added to the manuscript in section 3.1 with the relevance to the device. Section 2.2 consists of generalized equations for the potential and depletion width in any semiconductor channel. The band diagram has been corrected and redrawn for the manuscript. The positive charges/voltage on the SiO2-electrolyte interface due to protonation or applied gate bias accumulate negative charges near the Silicon-SiO2 interface that bend the energy bands downward keeping the Fermi level close to the conduction band as compared to the bulk region. The signs/definitions are corrected in the manuscript.

  • In section 2.3, the description of the FinFET would be redundant. It unfocuses the importance of the work. And also, the border of the numerical calculation and the experimental result is unclear. If any experimentals were performed, the details of that (including how the results were combined with calculations) should be described.

Comment: The FinFET device parameters are described to clarify the use of Gouy-Chapman-Stern and site-binding model. As the model describes the interface properties, it can be applied to any FET. We chose the FinFETs, because are the ones being fabricated in LIST, however, it also provides some additional advantages: The high-aspect-ratio of the preferred FinFET device is important for the factor to keep the results as linear as possible. The FinFET device behaves as 2 planar surfaces for the charge interactions excluding the device thickness. A thicker device may introduce non-linearity in the pH-sensing due to lower electrostatic control over the channel inducing a higher leakage current and parasitic passive elements. A higher dynamic range of the High-aspect-ratio FinFET allowed absolute pH sensing even with parametric variations. The experimental setup is well described in references 27 (S. Rollo, 2019) and 28 (S. Rollo, 2020). The variation of device characteristics with the change in pH can be detected easily but taking device degradation as a big factor in pH sensing clouds the accurate results. The long exposure of oxide to the electrolyte may provide the same device characteristics for different pH values. The uncertainty in the pH sensing due to oxide degradation prevents the detection of absolute pH value. Thus, this work solves the problem by using more than a single device to map the device characteristics for different degradation values across the pH range.

  • The reviewer recommend to think about the phrase of “absolute pH value”. Generally, the measurement of field-effect chemical sensors detect “pH changes”.

Comment: Indeed, it is true that FET chemical sensors detect “pH changes” but the detected pH value may not be right considering the result from a single device or an array of similar devices due to the strong dependence of device characteristics on parametric variations. Thus, the absolute pH value can only be known by overlapping the data from the different devices whose characteristics vary differently due to the oxide degradation. Thus, this work is focused on designing a methodology to implement an “absolute pH sensor”.

Minor:

  • The abstract would better explain “the paper is about IS-FET” (otherwise, more widely field-effect chemical sensors) for the readers.

Comment: The operation of the simulated device and analytical model is generalized for any IS-FET device. The calculation of the charge density with the effective interface potential calculations works with any charged bearing electrolyte. For the transducing action, IS-FET is a broader category that accompanies the FET chemical sensor. However, the same approach may not work for every chemical sensor (depending on several factors, affinity, type of reaction, bonds, catalyst, etc.) but every IS-FET can be modeled using the site-binding and Gouy-Chapman-Stern model.

  • The comparison of the dielectric constants and chemical stability of general materials for the gate passivation layer (such as Silicon nitride, Tantarum oxide, Aluminum oxide) with HfO2 would emphasize the importance

Comment: There are several works [references] that focused on the stability and chemical stability of general materials for different device architectures. The presented work also took into account the comparison of SiO2 and HfO2 as gate oxides. However, the lack of any work related to extracting the accurate pH value while considering the oxide degradation was the motivation for this work. The simulated device with HfO2 had a less pronounced effect of the oxide degradation that may have been the case of considering other oxides (such as Silicon nitride, Tantarum oxide, Aluminum oxide). Thus, to properly demonstrate the effect of oxide degradation, the designed methodology was followed.

  • Rollo, S., Rani, D., Olthuis, W. and García, C.P., 2020. High-performance Fin-FET electrochemical sensor with high-k dielectric materials. Sensors and Actuators B: Chemical, 303, p.127215.
  • Narang, R., Saxena, M. and Gupta, M., 2017. Analytical model of pH sensing characteristics of junctionless silicon on insulator ISFET. IEEE Transactions on Electron Devices, 64(4), pp.1742-1750.
  • Jang, H.J., Kim, M.S. and Cho, W.J., 2011. Development of Engineered Sensing Membranes for Field-Effect Ion-Sensitive Devices Based on Stacked High-$\kappa $ Dielectric Layers. IEEE electron device letters, 32(7), pp.973-975.
  • Lu, C.H., Hou, T.H. and Pan, T.M., 2017. High-Performance Double-Gate $\alpha $-InGaZnO ISFET pH Sensor Using a HfO2 Gate Dielectric. IEEE Transactions on Electron Devices, 65(1), pp.237-242.
  • The caption of Fig.6c – d begins as “Calculated external bias … as a function of the degradation…”, but the plot is actually converted in pH scale. It is not kind for readers.

Comment: As per your suggestion, the corrections have been made in the manuscript.

 The word of “mapping” is unclear.

Comment: Here, mapping is considered as a methodology for planning and assessing different results that aim to bring about ‘real’ and tangible common change. Bring the results of two different devices with the same input parameter (pH) providing the same output (current/voltage) for an unknown parameter (oxide degradation) is the mapping.